# Chemical Variation of Leaves and Pseudobulbs in *Prosthechea karwinskii* (Orchidaceae) in Oaxaca, Mexico

**DOI:** 10.3390/plants14040608

**Published:** 2025-02-18

**Authors:** Gabriela Soledad Barragán-Zarate, Beatriz Adriana Pérez-López, Manuel Cuéllar-Martínez, Rodolfo Solano, Luicita Lagunez-Rivera

**Affiliations:** Laboratorio de Extracción y Análisis de Productos Naturales Vegetales, Centro Interdisciplinario de Investigación para el Desarrollo Integral Regional Unidad Oaxaca, Instituto Politécnico Nacional, Hornos 1003, Santa Cruz Xoxocotlán C.P. 71230 Oaxaca, Mexico; gbarraganz@ipn.mx (G.S.B.-Z.); estudioorquideasdeoaxaca@gmail.com (B.A.P.L.); mcuellarm@ipn.mx (M.C.-M.)

**Keywords:** bioactive compounds, medicinal plants, extraction yield, UHPLC-ESI-qTOF-MS/MS, geographical variation

## Abstract

*Prosthechea karwinskii* is an endemic orchid of Mexico with significant value for its traditional uses: ornamental, ceremonial, and medicinal. The pharmacological activity of this plant has been studied using specimens recovered from religious use during Holy Week in Oaxaca, Mexico, sourced from various localities within this state. Geographical variability can influence the chemical composition of plants, as environmental factors affect the production of their secondary metabolites, which impact their biological properties. This research evaluated the variability in the chemical composition of leaves and pseudobulbs of *P. karwinskii* obtained from different localities in Oaxaca, comprising 95–790 g and 376–3900 g of fresh material for leaves and pseudobulbs, respectively, per locality. Compounds were identified using UHPLC-ESI-qTOF-MS/MS following ultrasound-assisted hydroethanolic extraction. Twenty-one compounds were identified in leaves and twenty in pseudobulb. The findings revealed differences in chemical composition across localities and between leaves and pseudobulbs of the species. The Roaguia locality exhibited the highest extraction yield and pharmacological potential in leaves. For pseudobulbs, Cieneguilla specimens showed the highest yield, and El Lazo had the lowest yield but the highest pharmacological potential. This study represents the first comprehensive analysis of the variation in the chemical composition of a native Mexican orchid. In all localities, leaves and pseudobulbs contained compounds with known biological activity, validating the use of the species in traditional medicine and highlighting its potential for medical and biological applications.

## 1. Introduction

The orchid *Prosthechea karwinskii* (Mart.) J.M.H. Shaw is endemic to the mountains of southern Mexico and is considered one of the most striking species of Mexican flora due to its ornamental and cultural value. In Oaxaca, this orchid is traditionally used for religious purposes during Holy Week celebrations and has applications in traditional medicine. The leaves are used to reduce glucose levels in individuals with diabetes; the pseudobulbs are used for healing wounds and burns, as well as for treating coughs and diabetes; and the flowers are used to prevent miscarriage, help in labor, and relieve coughs [1].

Previous studies have identified compounds in the leaves, pseudobulbs, and flowers of *P. karwinskii* that are associated with its medicinal properties. The leaves are rich in phenols and flavonoids, which contribute to the orchid’s antioxidant activity [2]. Extracts from leaves, pseudobulbs, and flowers have been shown to reduce body fat, glucose, total cholesterol, and triglyceride levels in Wistar rats with induced metabolic syndrome [3]. Furthermore, leaf extracts demonstrate antioxidant and anti-inflammatory properties, along with a gastroprotective effect against damage caused by non-steroidal anti-inflammatory drugs (NSAIDs) in Wistar rats [4]. These extracts also mitigate obesity, insulin resistance, inflammation, and cardiovascular risk in Wistar rats with induced metabolic syndrome [5]. Additionally, the extracts reduce oxidative stress by regulating the activity of antioxidant enzymes such as superoxide dismutase and catalase [6], offer cardiovascular protection [7], and have the potential for treating atherothrombosis [8].

The biological activity of *P. karwinskii* reported in these studies has been evaluated using extracts from specimens recovered after religious use in Zaachila, Oaxaca. These specimens were originally collected from their natural habitats for such purposes [9]. However, the potential effects of geographic variability on the chemical composition of this species were not considered. The content of secondary metabolites in plants is influenced by environmental factors that induce various types of stress. These stress conditions can be classified as abiotic or biotic. Abiotic stress factors include chemical or physical imbalance in the environment, temperature fluctuations, drought, salinity, light exposure, flooding, nutrient availability, altitude, and phytotoxins. In contrast, biotic stress factors include viroids, fungi, viruses, bacteria, nematodes, oomycetes, protists, mycoplasmas, invertebrates, and competing plants [10,11,12]. Plants respond to stress at morphological, anatomical, biochemical, and molecular levels [10], which influences the synthesis of secondary metabolites. These changes can, in turn, impact the quality and properties of plants used for medicinal purposes [10,11,12]. Additionally, different plant parts may produce distinct secondary metabolites [11].

Regarding orchids, environmental factors influence the chemical composition of *Epidendrum ciliare* L. floral fragrances, causing variation among localities [13]. Other studies have described variations in the colors, scents, nectar, and essential oils produced by orchid flowers [14,15,16,17]. However, in these plants, studies analyzing chemical composition variation in structures other than their flowers are scarce. A recent study demonstrated geographic variation in the floral morphology of *P. karwinskii* among different localities in Oaxaca, identifying traits that are useful for distinguishing local forms [18]. While populations of this species inhabit different regions with similar vegetation types, variations in elevation may influence environmental conditions. As previously mentioned, such variability can affect the chemical composition of plants. Therefore, in this study, we expect geographical differences (among localities) in the composition of the biocompounds present in the leaves and pseudobulbs of *P. karwinskii*. To explore this further, the objective of this research was to evaluate the variation in the chemical composition of the leaves and pseudobulbs of *P. karwinskii* sampled from six different localities representing its geographical distribution in Oaxaca, Mexico.

## 2. Results

### 2.1. Extraction Yields

Table 1 presents the extraction yields of leaves and pseudobulbs of *P. karwinskii* across the different localities. Significant differences in yield were observed both between plant parts and between localities, with leaves generally producing higher percentage yields. Among the leaf extracts, Roaguia showed the highest yield, while San Sebastian de las Grutas displayed the lowest. For pseudobulbs, the highest yield was recorded in Cieneguilla, and the lowest in El Lazo.

### 2.2. Compounds Detected by UHPLC-ESI-qTOF-MS/MS Analysis

Table 2 and Table 3 provide detailed information on the retention time (RT) of each compound, its *m*/*z* value, error, the fragments formed, tentative identification with the corresponding formula, and relative intensity in the extracts of leaves and pseudobulbs of *P. karwinskii* from the six evaluated localities.

Figure 1 provides comparative chromatograms of the hydroethanolic extracts from leaf (Figure 1A) and pseudobulb (Figure 1B) of *P. karwinskii*, respectively, collected from different localities in Oaxaca. While the same compounds were present in each plant part across all localities, variations in the relative intensity of the peaks were observed. This indicates that the concentration of secondary metabolites differs depending on the geographical origin of the plant material.

### 2.3. Heatmap Analysis of the Variation in Compounds in Leaves and Pseudobulbs of P. karwinskii

Figure 2A presents a heatmap showing the relative intensity of the compounds identified in the leaf extracts of *P. karwinskii* from six localities in Oaxaca, Mexico. The extract from Roaguia exhibited higher intensities of rutin, chlorogenic acid, uridine, guanosine, neochlorogenic acid, and D-tagatose. El Lazo’s extract displayed elevated levels of kaempferol-3-O-rutinoside, azelaic acid, sebacic acid, succinic acid, L-(-)-phenylalanine, embelin, 9,12,13-trihydroxy-10(E),15(Z)-octadecadienoic acid, pinelic acid, eicosenoic acid, 9-hydroperoxy-10E,12Z-octadecadienoic acid, and 12,13-epoxy-9Z-octadecadienoic acid. Cieneguilla extracts showed the highest intensities for quinic acid and malic acid. Meanwhile, the extracts from El Molino and Amialtepec exhibited the highest intensities for isocitric acid and gibberellin A7, respectively. According to the Heatmap colorations, the compound found at a higher intensity in *P. karwinskii* leaves is kaempferol-3-O-rutinoside.

Figure 2B presents a heatmap showing the relative intensity of compounds identified in the extracts of *P. karwinskii* pseudobulbs from six different localities in Oaxaca. The extract from Cieneguilla exhibited higher intensities of D-tagatose and feruloyltyramine. In contrast, the extract from El Molino showed greater intensities of malic acid, isocitric acid, diosmin, embelin, and 12,13-epoxy-9Z-octadecenoic acid. The extract from Roaguia displayed the highest intensities of succinic acid and 3-methylglutaric acid. El Lazo’s extract demonstrated the highest intensities for deoxyloganic acid, neochlorogenic acid, chlorogenic acid, rutin, azelaic acid, sebacic acid, pinelic acid, and gigantol. Meanwhile, the extract from Amialtepec had the highest intensities of quinic acid, 9-hydroperoxy-10E,12Z-octadecadienoic acid, and 13-hydroperoxyoctadeca-9,11-dienoic acid. According to the Heatmap staining, the compound found at a higher intensity in the pseudobulbs of *P. karwinskii* is 13-hydroperoxyoctadeca-9,11-dienoic acid.

Figure 3 presents a comparative heatmap showing the relative intensity of compounds identified in the leaf and pseudobulb extracts of *P. karwinskii* across six localities. In the leaf extracts, 21 compounds were detected, seven of which were unique to this part of the plant: uridine, L-(-)-phenylalanine, guanosine, kaempferol-3-O-rutinoside, 9,12,13-trihydroxy-10(E),15(Z)-octadecadienoic acid, eicosenoic acid, and 9-hydroperoxy-10E,12Z-octadecadienoic acid. Conversely, 20 compounds were identified in the pseudobulb extracts, six of which were unique to this part of the plant: 3-methylglutaric acid, deoxyloganic acid, diosmin, feruloyltyramine, gigantol, and 13-hydroperoxyoctadeca-9,11-dienoic acid. Fifteen compounds were common to both plant parts, though their intensities varied depending on both the plant part and the locality.

Figure 4 shows a dendrogram illustrating the clustering of detected compounds in the leaf and pseudobulb extracts of *P. karwinskii* from six localities. The clustering is determined by the plant part from which the extracts were obtained: pseudobulb extracts are grouped separately from leaf extracts.

Within the pseudobulb group, two subgroups are observed. One subgroup includes the localities in southern Oaxaca (El Lazo and Amialtepec), while the other comprises Roaguia (Valles Centrales), San Sebastian de las Grutas (southern Oaxaca), and the Mixteca localities (El Molino and Cieneguilla). The grouping of leaf extracts reveals a different pattern. Cieneguilla forms a subgroup with El Lazo, San Sebastian de las Grutas, and Amialtepec, while another subgroup consists solely of El Molino and Roaguia.

Table 4 highlights the compounds found in the leaves and pseudobulbs of *P. karwinskii* that are associated with biological activities relevant to the medicinal use of this species, including their ability to improve glucose metabolism, anti-inflammatory properties, and antioxidant activity. The table also indicates the sampled locality where each compound is found in the highest intensity.

## 3. Discussion

According to Cruz-García et al. [1], both the leaves and pseudobulbs of *P. karwinskii* are used in traditional medicine for treating diabetes. Additionally, pseudobulbs are utilized to treat coughs, as well as wounds and burns. These medicinal uses are linked to inflammatory processes. For example, respiratory issues are associated with bronchial inflammation [33]; wound and burn healing involves various types of inflammatory cells [62]; and both inflammation and oxidative stress are implicated in the pathophysiology of diabetes [63]. Furthermore, both plant parts exhibit a similar ability to inhibit reactive oxygen species (ROS) [2], which could be related to the similarity in their chemical profiles. Elevated glucose levels in diabetes are known to increase ROS production [64], and excessive ROS can lead to oxidative stress. Inflammation also contributes to oxidative stress, and in turn, oxidative stress exacerbates inflammation [41]. Thus, the plant’s effects on ROS and oxidative stress may be linked to its traditional medicinal uses.

Since the traditional medicinal uses of *P. karwinskii* leaves and pseudobulbs are related to inflammation, oxidative stress, and glucose metabolism disorders [1,2], and both plant parts contain compounds with these biological activities (Table 4), it can be inferred that extracts from localities where these compounds are more abundant may have greater pharmacological potential.

Environmental conditions play a crucial role in determining the concentration of secondary metabolites in medicinal plants, as plants produce specific types and amounts of metabolites necessary to counteract the environmental stress they experience [11]. According to Sampaio et al. [65], drought stress can reduce photosynthetic activity, increasing ROS production and stimulating the production of phenolic compounds as a defense mechanism. Heat stress during drought can affect metabolic regulation, water permeability, and CO_2_ levels, thereby enhancing antioxidant properties and increasing carbohydrate availability. Additionally, UV-B radiation stress can lead to the production of phenolic compounds that absorb and/or dissipate solar energy, helping to prevent the formation of free radicals and other oxidative species. Phenolic compounds, such as flavonoids, accumulate in plants in response to abiotic stress and are considered a fundamental mechanism for the elimination of ROS [10].

The review by Punetha et al. [10], analyzing the effect of abiotic stress on secondary metabolite production in medicinal plants, indicates that phenolic acids and flavonoids are enhanced by increased UV radiation, drought, high temperatures, and salinity. In addition, the review by Pant et al. [11] found that these compounds are promoted by high temperatures, elevated concentrations of carbon dioxide and ozone, and UV radiation. Additionally, rutin and chlorogenic acid, compounds identified in *P. karwinskii*, are favored by higher light intensity.

Sun et al. [66] compared the metabolites of Lushan Yunwu tea leaves from different geographical regions, linking the observed variations to altitude. At higher altitudes, diffuse light promotes nitrogen metabolism, amino acid production, and the synthesis of nitrogen compounds, while increasing the content of organic acids and flavonoids, though total phenol content decreases. Our results did not show a similar pattern for *P. karwinskii* extracts. However, chlorogenic acid was found in higher concentrations in the leaf extract from the highest altitude locality (Cieneguilla), but not in the pseudobulb extract. Conversely, rutin content was higher in the pseudobulb extract from the lower-altitude locality (San Sebastian de las Grutas), but not in the leaf extract.

In contrast to the previous analysis [2] of Soxhlet extracts of leaves and pseudobulbs of *P. karwinskii* collected from churches in Zaachila Oaxaca, in the present study, more compounds were identified for each part of the plant. This variation could be due to the method used for the extraction of the compounds, as well as to the loss or degradation of compounds due to the time they remained on the altars of the churches because of their ceremonial use and the manipulation of the plants. In the previous study, the origin of the plant was not considered, as they had been collected from different parts of the state for ceremonial purposes.

This study represents the first analysis of chemical composition variations in the orchid *P. karwinskii*, as well as other Mexican orchid species from different localities. The findings highlight the need for continued research into how these chemical variations correlate with physical and environmental factors of the localities where the orchid is found. The results presented here, along with those obtained in previous studies on *P. karwinskii*, lead to questions for future research. The habitat in each of the sampling localities for *P. karwinskii* is very similar, with this epiphytic orchid growing on the same host trees, *Quercus* sp. Could the variation in chemical composition detected among the species’ localities be related to the environmental factors that characterize them? Several compounds identified in the leaves and pseudobulbs of *P. karwinskii* are known for their anti-inflammatory, antioxidant, and glucose metabolism-regulating activities. When extracting compounds from these localities, could any of them show better performance in evaluating these activities using an *in vivo* model?

It is also interesting to compare these results with those of Santos-Escamilla et al. [18], who documented variations in the floral morphology of *P. karwinskii* across different populations in the state of Oaxaca, where Albarradas presented the most differentiated population of *P. karwinskii*. Notably, the leaf extract from the locality of Roaguia, in the municipality of San Lorenzo Albarradas, contained a higher intensity of compounds with reported biological activity associated with the plant’s traditional uses (Table 4). This suggests a possible link between floral morphological variation and chemical composition in this species. Could the compounds identified here serve as chemical markers to recognize intraspecific variation in *P. karwinskii*?

## 4. Materials and Methods

### 4.1. Plant Material

Plant material was collected from six localities representing the distribution of *P. karwinskii* in Oaxaca, Mexico. For this, a scientific collector’s permit was obtained from the Mexican Ministry of Environment and Natural Resources (00851/06 and 02228/17). Table 1 provides information on these localities, while the map in Figure 5 indicates its geographic location. Sampling was conducted from January to February 2018. From each adult plant (>15 pseudobulbs), only divisions containing 2–3 pseudobulbs with their leaves were collected, comprising 95–790 g and 376–3900 g of fresh material for leaves and pseudobulbs, respectively, per locality, leaving the remaining parts of the plant on its host tree to ensure its survival. Additionally, a voucher specimen was collected from each locality, herborized, and deposited in the OAX Herbarium of Instituto Politécnico Nacional (see Table 1). The taxonomic identity of the species was validated by one of the authors (R.S.), following Villasenor’s checklist in the assigned scientific name [67].

### 4.2. Conditioning of Plant Material and Obtaining Extracts

The plant material from each locality was separated into leaves and pseudobulbs. Each part was individually weighed, washed with water, and dried at 50 °C. After drying, the material was ground and sieved (Figure 6). Ultrasound-assisted extraction was performed on the pulverized leaf and pseudobulb samples from each locality, following the method described by Barragán-Zárate et al. [4]. The extraction was carried out using an ultrasonic processor (VCX 750, Cientifica Senna, Mexico City, Mexico) set at a frequency of 20 kHz and an amplitude of 30%. A 50% (*v*/*v*) ethanol–water mixture was used as the solvent, with a sample-to-solvent ratio of 1 g:18 mL. The extraction was performed at 40 °C for 20 min, with cycles of 10 s of operation followed by 5 s of rest.

### 4.3. Compound Identification by UHPLC-ESI-qTOF-MS/MS

To determine the compound profile of the leaves and pseudobulbs from each locality, the method described by Barragán-Zárate et al. [2] was followed. One mg of each extract was analyzed using an ultra-high-performance liquid chromatography (UHPLC) system (Thermo Scientific Ultimate 3000, Waltham, MA, USA) coupled with an Impact II mass spectrometer (Bruker, Billerica, MA, USA). The mass spectrometer operated with electrospray ionization (ESI) and quadrupole time-of-flight (qTOF) detection. The analysis was performed using a Thermo Scientific Acclaim 120 C18 column (2.2 μm, 120 Å, 50 × 2.1 mm). The mobile phase consisted of: (A) 0.1% formic acid in water and (B) acetonitrile. The gradient elution program was as follows: 0% B (0–2 min), 1% B (2–3 min), 3% B (3–4 min), 32% B (4–5 min), 36% B (5–6 min), 40% B (6–8 min), 45% B (8–9 min), 80% B (9–11 min), and 0% B (12–14 min). The injection temperature was 25 °C, and the flow rate was set at 0.35 mL/min. The analysis was conducted in negative electrospray ionization mode at 0.4 bar (5.8 psi) with autoMSMS, in the mass range 50–700 *m*/*z*. The ionization of the capillary voltage (Vcap) was 2700 V. Data obtained from the UHPLC-ESI-qTOF-MS/MS analysis were processed using DataAnalysis software 3.1 (Bruker) and imported into MetaboScape 3.0 (Bruker) for further analysis, including peak extraction. For each compound, retention time (RT), *m*/*z*, fragmentation patterns, and peak intensities were recorded. Compounds were identified by comparing their exact mass and MS/MS spectra with those in the MetaboBase 3.0 (Bruker) and MassBank libraries, as well as by consulting relevant scientific literature.

### 4.4. Statistical Analysis

The heatmap is a visualization that simultaneously reveals row and column hierarchical cluster structure in a data matrix [68]. Each heatmap was generated based on the relative intensity values of the peaks from the analyses [69,70] to visualize differences in chemical compounds between leaves and pseudobulbs, as well as across localities. The heatmaps were created with the Pheatmap 1.0.12 package [71] with complete linkage as the agglomeration method. Additionally, to explore the chemical composition dataset and assess whether it could be summarized into several clusters that were similar within themselves and distinct from others, a cluster analysis was conducted.

The hierarchical cluster analysis was performed with the hclust function of the stats package, with the complete linkage as the agglomeration method, where the distance or similarity between clusters occurs by the maximum distance or minimal similarity among its components [72]. Both analyses were carried out using RStudio 2023.03.0 [73] and using Euclidian distance.

## 5. Conclusions

*Prosthechea karwinskii* exhibits variation in the chemical composition of extracts derived from its leaves and pseudobulbs, as well as differences based on the locality of origin of the plant material. The chemical composition is more consistent among localities for the same plant part than between leaves and pseudobulbs from the same locality. Leaf extracts from El Lazo and Roaguia showed higher intensities of compounds previously reported to have biological activities associated with the medicinal use of the species. Similarly, pseudobulb extracts from El Lazo also exhibited a higher intensity of these bioactive compounds. Extracts from these localities demonstrate greater potential for addressing health issues related to glucose metabolism, inflammation, and oxidative stress. In terms of extraction yield, Roaguia produced the highest yields for both leaves and pseudobulbs, while El Lazo had the lowest yield for pseudobulbs. The findings of this study raise questions for future research on *P. karwinskii*. Despite growing in similar habitats, variations in chemical composition among localities may be influenced by environmental factors. Additionally, some identified compounds have bioactive properties, prompting the question of whether extracts from certain localities may perform better within *in vivo* models. Finally, these compounds could serve as chemical markers for intraspecific variation in *P. karwinskii.*

## Figures and Tables

**Figure 1 plants-14-00608-f001:**
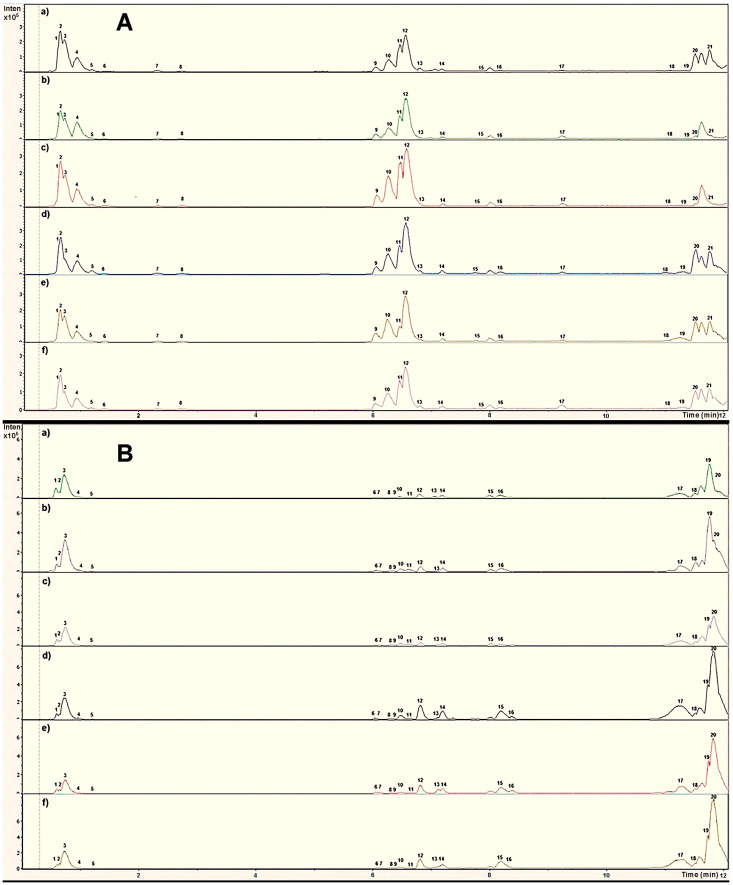
Chromatograms obtained with UHPLC-ESI-qTOF-MS/MS from the leaf (**A**) and pseudobulb (**B**) extracts of *Prosthechea karwinskii* from six localities of Oaxaca, Mexico. (**a**) Cieneguilla, Santo Domingo Yanhuitlan, (**b**) El Molino, San Pedro y San Pablo Teposcolula, (**c**) Roaguia, San Lorenzo Albarradas; (**d**) El Lazo, San Miguel Sola de Vega, (**e**) San Sebastian de las Grutas, San Miguel Sola de Vega, (**f**) Amialtepec, Santa Catarina Juquila.

**Figure 2 plants-14-00608-f002:**
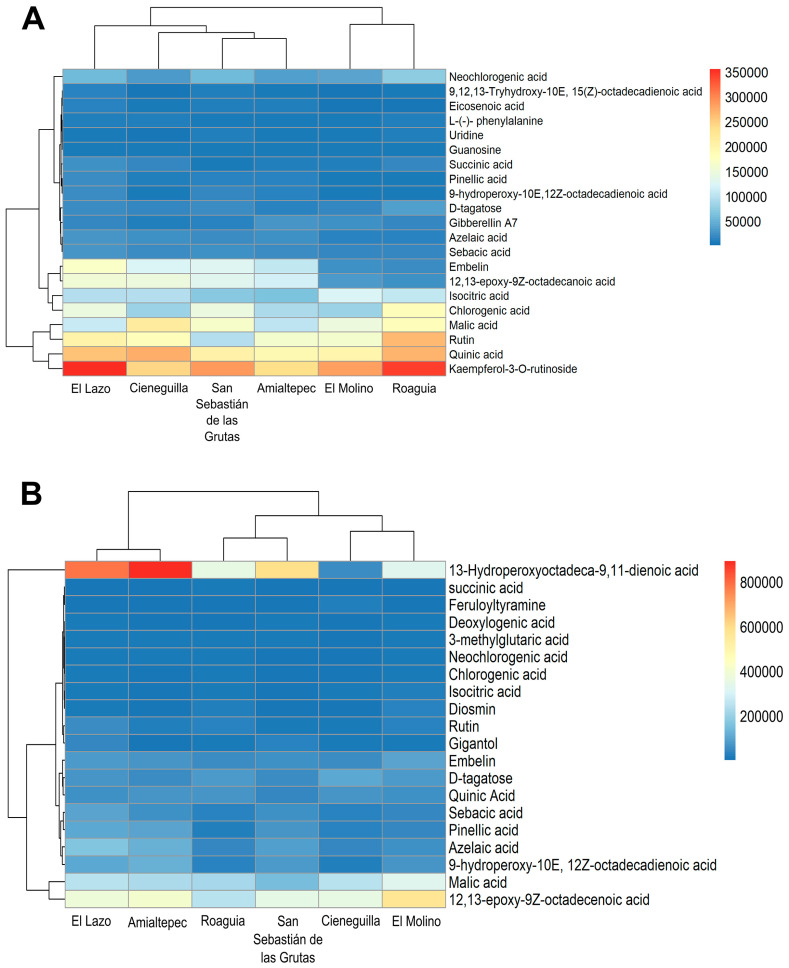
Heatmap showing the relative intensity of compounds identified in the leaf (**A**) and pseudobulb (**B**) extracts of *Prosthechea karwinskii* from six localities of Oaxaca, Mexico. The relative intensity scale on the right ranges from red, indicating the highest values, to blue, indicating the lowest.

**Figure 3 plants-14-00608-f003:**
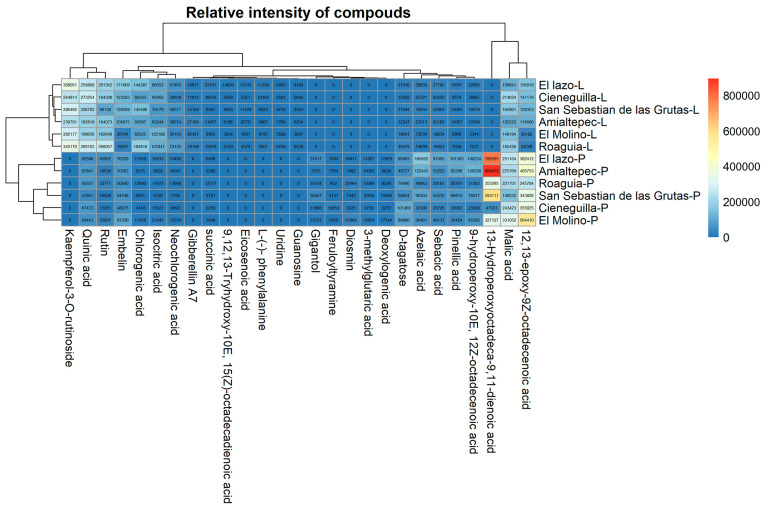
Heatmap showing the relative intensity of compounds identified in the leaf (L) and pseudobulb (P) extracts of *Prosthechea karwinskii* from six localities in Oaxaca, Mexico. Each cell value represents the relative intensity of a compound. The scale on the right indicates relative intensity, with red denoting the highest values and blue the lowest.

**Figure 4 plants-14-00608-f004:**
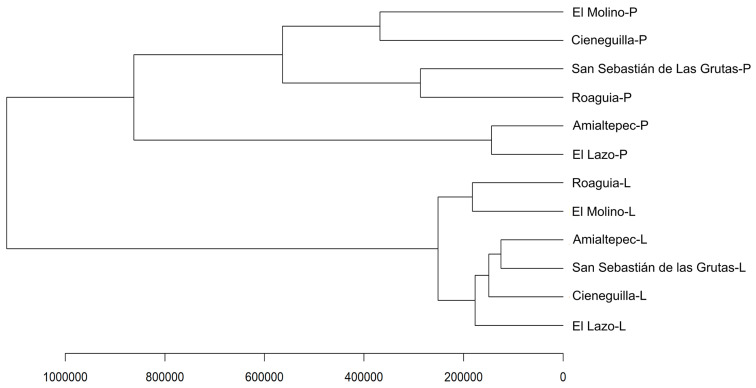
Dendrogram generated using a hierarchical clustering algorithm and clustering index, illustrating the grouping of localities based on the detected chemical compounds in the leaf (L) and pseudobulb (P) extracts of *Prosthechea karwinskii*.

**Figure 5 plants-14-00608-f005:**
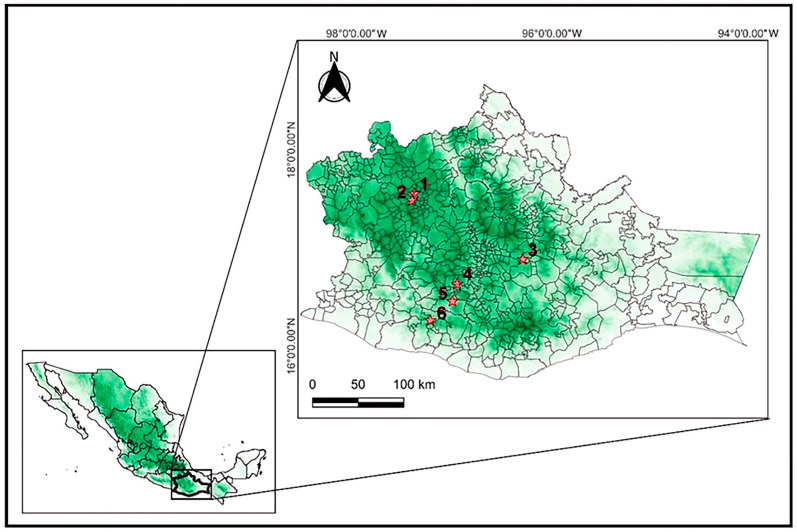
Map of the six localities of *Prosthechea karwinskii* in Oaxaca, Mexico sampled in this study. 1: Cieneguilla, Santo Domingo Yanhuitlan, 2: El Molino, San Pedro y San Pablo Teposcolula, 3: Roaguia, San Lorenzo Albarradas, 4: El Lazo, San Miguel Sola de Vega, 5: San Sebastian de las Grutas, San Miguel Sola de Vega, 6: Amialtepec, Santa Catarina Juquila.

**Figure 6 plants-14-00608-f006:**
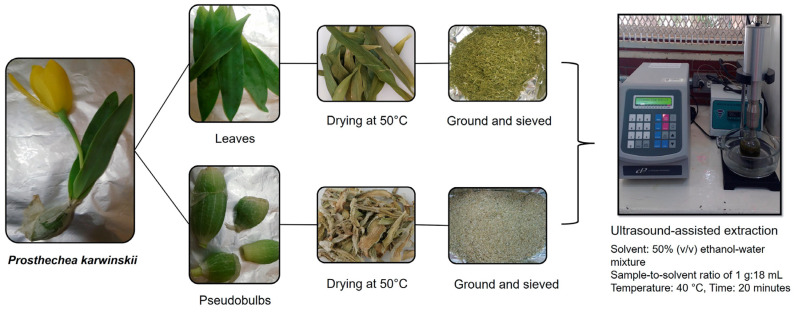
Process scheme for the extraction of compounds from leaves and pseudobulbs of *Prosthechea karwinskii*.

**Table 1 plants-14-00608-t001:** Extraction yields of leaves and pseudobulbs of *Prosthechea karwinskii* collected in six localities in the state of Oaxaca, Mexico. Yield values are reported as the mean ± standard deviation of three replicates.

Locality, Municipality	Voucher	Altitude	Habitat	Plant Part	Yield (%)
Cieneguilla, Santo Domingo Yanhuitlan	Solano 4246 OAX	2409	Oak-Pine Forest	Leaves	24.3 ± 5.0
				pseudobulbs	23.3 ± 3.0
El Molino, San Pedro y San Pablo Teposcolula	Solano s.n. OAX	2411	Oak-Pine Forest	Leaves	25.0 ± 3.6
				pseudobulbs	14.0 ± 1.0
Roaguia, San Lorenzo Albarradas	Solano 4441 OAX	2450	Oak-Pine Forest	Leaves	30.0 ± 6.0
				pseudobulbs	22.0 ± 4.3
San Sebastian de las Grutas, San Miguel Sola de Vega	Solano 4334 OAX	1720	Oak Forest	Leaves	18.0 ± 6.4
				pseudobulbs	15.0 ± 1.0
El Lazo, San Miguel Sola de Vega	Solano s.n. OAX	1840	Oak Forest	Leaves	21.6 ± 1.5
				pseudobulbs	12.0 ± 1.0
Amialtepec, Santa Catarina Juquila	Solano 1806 OAX	2105	Oak Forest	Leaves	22.3 ± 7.3
				pseudobulbs	19.3 ± 9.0

**Table 2 plants-14-00608-t002:** Summary of the values for the compounds detected by UHPLC-ESI-qTOF-MS/MS in the leaf extracts of *Prosthechea karwinskii* from six localities in Oaxaca, Mexico. PN: Peak number, RT: Retention Time. CIE: Cieneguilla, Santo Domingo Yanhuitlan. EMO: El Molino, San Pedro y San Pablo Teposcolula. ROA: Roaguia, San Lorenzo Albarradas. ELA: El Lazo, San Miguel Sola de Vega. SSG: San Sebastian de las Grutas, San Miguel Sola de Vega. AMI: Amialtepec, Santa Catarina Juquila.

PN	RT	*m*/*z* (M-H)	Error (ppm)	Fragments	Compound (Chemical Formula) *	Relative Intensity					
						CIE	EMO	ROA	ELA	SSG	AMI
1	0.6	179.0555	2.2	89.0225, 101.0233	D-tagatose ^a,b^ (C_6_H_12_O_6)_	15,932	19,041	35,079	21,016	21,294	12,347
2	0.7	191.0557	1.3	85.0293, 87.0078, 111.0443, 127.6945	Quinic Acid ^b^, [19,20] (C_7_H_12_O_6_)	273,254	198,656	269,103	258,808	206,762	192,818
3	0.8	133.0140	0.6	115.0032	Malic acid ^b^, [20,21,22] (C_4_H_6_O_5)_	214,624	148,194	185,426	109,841	165,901	102,223
4	1.1	191.0189	1.3	85.0290	Isocitric acid ^b^ (C_6_H_8_0_7_)	95,658	122,188	107,847	96,552	70,179	65,244
5	1.2	117.0191	1.1	73.0290, 99.0072	Succinic acid ^b^, [23] (C_4_H_6_O_4_)	16,916	9855	15,878	23,167	8367	11,657
6	1.4	243.0609	2.4	110.0247	Uridine ^b^ (C_9_H_12_N_2_O_6_)	4501	5508	10,765	6992	8776	7766
7	2.3	164.0712	1.6	72.0072, 103.0539, 147.0442	L-(-)-Phenylalanine ^a,b^ (C_9_H_11_NO_2_)	10,384	6797	9201	11,228	6829	6067
8	2.7	282.0833	5.7	108.5347, 133.0157, 150.0429	Guanosine ^a,b^, [23] (C_10_H_13_N_5_O_5)_	5940	5697	8539	8199	6524	6254
9	6.0	353.0867	3.4	173.0430, 179.0365, 191.0556	Neochlorogenic acid ^b^, [19,23] (C_16_H_18_O_9)_	32,839	39,103	73,155	57,463	58,317	36,874
10	6.3	353.0866	3.3	173.0452, 179.0365, 191.0556	Chlorogenic acid ^b^, [24] (C_16_H_18_O_9)_	82,543	82,525	184,816	144,387	145,486	92,587
11	6.5	609.1438	2.4	300.0266, 301.0335	Rutin ^a,b^, [19,24,25,26] (C_27_H_30_O_16_)	184,308	162,646	266,057	201,362	96,138	164,073
12	6.6	593.1489	3.1	284.0314, 285.0393	Kaempferol-3-O-rutinoside ^a,b^ (C_27_H_30_O_15_)	244,814	282,177	345,170	356,051	286,492	236,781
13	6.8	187.0970	2.2	97.0653, 125.0963, 169.0889	Azelaic acid ^b^, [27] (C_9_H_16_O_4_)	25,721	13,536	14,839	28,839	19,554	23,313
14	7.2	201.1127	3.7	139.1128, 183.1021	Sebacic acid ^b^, [19] (C_10_H_18_O_4_)	20,350	18,634	19,024	27,785	22,903	22,182
15	7.8	327.2157	4.5	171.1023	9,12,13-Trihydroxy-10(E),15(Z)-octadecadienoic acid [19] (C_18_H_32_O_5_)	4569	3034	6130	14,026	9583	6106
16	8.2	329.2321	2.5	171.1023, 229.1436	Pinellic acid [27,28] (C_18_H_34_O_5_)	9574	5869	7558	19,591	13,093	15,067
17	9.2	329.1383	3.9	179.0717	Gibberellin A7 ^b^ (C_19_H_22_O_5_)	11,912	26,421	19,198	16,877	14,109	27,180
18	11.0	309.2055		291.1951 197.1187	Eicosenoic acid ^b^, [29] (C_20_H_38_O_2_)	5351	1621	4374	13,745	11,528	8779
19	11.3	311.2215	4.1	293.2102 171.1012 185.1190 197.1171	9-hydroperoxy-10E,12Z-octadecadienoic acid ^b^	6665	5344	7237	22,682	19,019	12,549
20	11.5	293.2112	5.6	223.1685, 235.1680, 275.2013	Embelin [27] (C_17_H_26_O_4_)	123,583	26,104	19,527	171,609	128,598	105,675
21	11.8	295.2270	3.8	277.2161 278.2182 183.1384 195.1375	12,13-epoxy-9Z-octadecenoic acid ^b^ (C_18_H_32_O_3_)	147,178	30,182	24,249	156,918	129,353	115,606

* Superscripts correspond to the sources included in the References section used in the identification of the compounds, except for ^a^, which was obtained from MetaboBase, and ^b^, which was obtained from MassBank.

**Table 3 plants-14-00608-t003:** Summary of the values for the compounds detected by UHPLC-ESI-qTOF-MS/MS in the pseudobulb extracts of *Prosthechea karwinskii* from six localities in Oaxaca, Mexico. PN: Peak number, RT: Retention Time. CIE: Cieneguilla, Santo Domingo Yanhuitlan. EMO: El Molino, San Pedro y San Pablo Teposcolula. ROA: Roaguia, San Lorenzo Albarradas. ELA: El Lazo, San Miguel Sola de Vega. SSG: San Sebastian de las Grutas, San Miguel Sola de Vega. AMI: Amialtepec, Santa Catarina Juquila.

PN	RT	*m*/*z* (M-H)	Error (ppm)	Fragments	Compound (Chemical Formula) *	Relative Intensity					
						CIE	EMO	ROA	ELA	SSG	AMI
1	0.6	179.0554	5.4	89.0225, 101.0233	D-tagatose ^a,b,c^ (C_6_H_11_O_6_)	101,469	80,866	76,466	69,493	50,654	49,727
2	0.7	191.0552	4.9	85.0293, 127.0396	Quinic acid ^b^, [19,20] (C_7_H_11_O_6_)	67,473	59,443	65,657	62,586	41,841	67,841
3	0.8	133.0142	4.7	115.0032	Malic acid ^b^, [20,21,22] (C_4_H_5_O_5_)	243,423	331,052	221,701	251,164	148,232	225,769
4	1.1	191.0189	1.3	85.0290	Isocitric acid ^b^ (C_6_H_8_O_7_)	15,627	21,045	11,672	18,533	5750	9828
5	1.2	117.0188	4.3	73.0290, 99.0072	Succinic acid ^b^, [23] (C_4_H_6_O_4_)	9209	7648	12,577	6499	2721	2200
6	6.1	145.0499	3.6	83.0491, 101.0606	3-Methylglutaric acid ^a^ (C_6_H_10_O_4_)	9759	12,854	13,009	11,827	12,204	10,093
7	6.1	359.0967	8.9	153.0550 197.0452 198.0479	Deoxyloganic acid ^b^ (C_16_H_24_O_9_)	5272	17,344	8526	17,829	15,064	6639
8	6.3	353.0861	3.4	173.0430, 179.0365, 191.0556	Neoclorogénic acid ^b^, [19,23] (C_16_H_18_O_9_)	6892	12,576	13,998	15,489	7185	10,367
9	6.4	353.0854	5.6	173.0452, 179.0365, 191.0556	Clorogénic acid ^b^, [24] (C_16_H_18_O_9_)	4149	11,438	12,090	13,169	6001	9235
10	6.5	609.1433	5.6	300.0266, 301.0335	Rutin ^a,b^, [19,24,25,26] (C_27_H_30_O_16_)	15,281	32,907	32,777	45,907	16,529	19,530
11	6.6	607.1272		299.0148 300.0248 301.0338 607.1246 608.1348 609.1458	Diosmin ^b^, [30] (C_28_H_32_O_15_)	9326	27,964	22,464	16,871	7442	7482
12	6.8	187.0967	3.5	97.0653, 125.0963, 169.0889	Azelaic acid ^b^, [27] (C_9_H_16_O_4_)	42,386	56,461	40,943	165,652	90,303	122,443
13	7.1	312.1231	2.2	148.0526 178.0502 297.0991	Feruloyltyramine ^b^ (C_18_H_19_NO_4_)	18,859	7459	952	1344	3131	7354
14	7.2	201.1125	2.8	139.1128, 183.1021	Sebacic acid ^b^, [27] (C_10_H_18_O_4_)	29,726	40,137	28,181	97,495	55,532	55,222
15	8.2	329.2315	4.7	171.1023, 229.1436	Pinellic acid [27,28] (C_18_H_34_O_5_)	32,062	39,424	20,701	101,363	66,913	92,206
16	8.4	273.1119	3.6	121.0297 98.0359 137.0624	Gigantol [31] (C_16_H_18_O_4_)	13,688	10,737	16,754	37,417	36,557	9725
17	11.3	311.2215	4.1	293.2102 171.1012 185.1190 197.1171	9-hydroperoxy-10E,12Z-octadecadienoic acid ^b^ (C_18_H_32_O_4_)	23,380	65,305	31,562	100,234	76,017	120,398
18	11.5	293.2112	5.6	223.1685, 235.1680, 275.2013	Embelin [27] (C_17_H_26_O_4_)	48,675	97,200	50,540	76,229	54,746	70,383
19	11.8	295.2270	3.8	277.2161 278.2182 183.1384 195.1375	12,13-epoxy-9Z-octadecenoic acid ^b^ (C_18_H_32_O_3_)	355,025	564,410	245,784	382,412	343,826	405,753
20	11.9	311.2212	4.1	293.2102 113.0966 195.1376	13-Hydroperoxyoctadeca-9,11-dienoic acid ^b^ (C_18_H_32_O_4_)	47,923	327,127	353,295	780,595	593,111	894,679

* Superscripts correspond to the sources included in the References section used in the identification of the compounds, except for ^a^, which was obtained from MetaboBase, ^b^, which was obtained from MassBank, and ^c^, which was obtained from Plant metabolites.

**Table 4 plants-14-00608-t004:** Compounds with biological activities related to the medicinal use of leaves and pseudobulbs of *Prosthechea karwinskii* collected from different localities in Oaxaca, Mexico. (-) means that the compound is not present in that part of the plant.

Compound	Type of Compound	Locality with Higher Relative Intensity		Biological Activity (Reference)
		Leaves	Pseudobulbs	
Quinic Acid	Carboxylic acid, cyclitol	Cieneguilla	Amialtepec	Anti-inflammatory [32]
Uridine	Pyrimidine nucleoside	Roaguia	-	Anti-inflammatory [33]
Guanosine	Purine nucleoside	Roaguia	-	Anti-inflammatory [33] Antioxidant [34]
Neochlorogenic acid	Phenolic acid	Roaguia	El Lazo	Anti-inflammatory [35,36] Anti-inflammatory and Antioxidant [37]
Chlorogenic acid	Phenolic acid	Roaguia	El Lazo	Improves glucose metabolism [38,39] Anti-inflammatory [35,36,40] Antioxidant [41,42]
Rutin	Flavonoid glycoside	Roaguia	El Lazo	Improves glucose metabolism [43] Antioxidant and Anti-inflammatory [44,45] Antioxidant [46]
Kaempferol-3-O-rutinoside	Flavonoid glycoside	El Lazo	-	Antioxidant [47] Anti-inflammatory [48]
Azelaic acid	Dicarboxylic acid	El Lazo	El Lazo	Antioxidant [49,50] Anti-inflammatory [51]
Pinellic acid	Trihydroxyoctadecenoic acid	El Lazo	El Lazo	Anti-inflammatory [52]
Embelin	Benzoquinone	El Lazo	El Molino	Improves glucose metabolism [53] Anti-inflammatory [54] Antioxidant and Anti-inflammatory [55]
Diosmin	Flavonoid glycoside	-	El Molino	Improves glucose metabolism and Anti-inflammatory [56] Antioxidant and Anti-inflammatory [57,58] Improves glucose metabolism, Anti-inflammatory and Antioxidant [59]
Gigantol	Bibenzyl	-	El Lazo	Anti-inflammatory [60,61]

## Data Availability

Data are contained within the article.

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
