# Peer review of "Chemical Variation of Leaves and Pseudobulbs in Prosthechea karwinskii (Orchidaceae) in Oaxaca, Mexico"

_plants, 2025, doi:10.3390/plants14040608_

Round 1

Reviewer 1 Report

Comments and Suggestions for Authors

Review of the manuscript ˝Variability in the chemical composition of leaves and pseudo-bulbs among localities of Prosthechea karwinskii (Orchidaceae) in Oaxaca, Mexico ˝ submitted to the Plants journal, section Phytochemistry, special issue Bioactivities of Nature Products.

The topic of the scientific work is interesting and original and contributes to the knowledge about the so far poorly researched endemic species Prosthechea karwinskii.

The abstract is written clearly highlighting the important objectives of the scientific paper.

The introductory part provides a clear overview of previous research and provides insight into the objectives of the work.

The research methodology is described in detail.

The results are clearly presented and well discussed, supported by relevant references.

Prior to publication I suggest the authors make minor corrections:

Line 79-80 – please correct to Figure 1A and 1B (not 2A and 2B)

Line 100 – please correct 9-acid,12,13-trihydroxy-10(E),15(Z)-octadecadienoic acid to 9,12,13-trihydroxy-10(E),15(Z)-octadecadienoic acid

Line 113 – according to the heatmap 2B Amialtepec had the highest intensities of gibberellin A7 (quinic acid is marked yellow implying that it is presented in lower concentrations). However, this compound is not listed in Table 3. Please check.

I would recommend the authors to put Tables 2 and 3 prior to the heatmaps.

Figure 3 should be replaced with a better resolution figure.

Table 2 – replace Uridin with Uridine

Figure 4 should be replaced with a better resolution figure.

In the Discussion part the authors could compare their results obtained by UHPLC with those of their previous work (ref. 2)

Author Response

RESPONSES TO REVIEWER’S COMMENTS

On behalf of all the authors, we sincerely appreciate the time and effort dedicated to reviewing this manuscript. Your feedback was invaluable in enhancing its quality. We have carefully considered your comments and made the corresponding revisions in the corrected manuscript. In the corrected version, the changes are highlighted in yellow, allowing you to see the modifications made. Additionally, we have provided a point.by- point response to each comment.

REVIEWER 1

Comments 1: [Line 79-80 – please correct to Figure 1A and 1B (not 2A and 2B).]

Response 1: [Thank you very much for your comments, the corresponding correction has been made (Lines 95 and 96).]

Comments 2: [Line 100 – please correct 9-acid,12,13-trihydroxy-10I,15(Z)-octadecadienoic acid to 9,12,13-trihydroxy-10I,15(Z)-octadecadienoic acid.]

Response 2: [Thank you very much for your comments, the name has been corrected. The change was made on line 114 of the manuscript.]

Comments 3: [Line 113 – according to the heatmap 2B Amialtepec had the highest intensities of gibberellin A7 (quinic acid is marked yellow implying that it is presented in lower concentrations). However, this compound is not listed in Table 3. Please check.]

Response 3: [We thank you very much for your observation, it is a detail that we missed when working on Figure 2 and Table 3, this detail of image 2 has been corrected. As for the difference in intensity according to the coloration, this was given for each specific compound and not in a general way, which was a little confusing. However, when looking at Figure 1B corresponding to the chromatogram of the pseudubulbs, at minute 9.2 (corresponding to gibberellin A7, no peak is observed). Figure 2 was redone so that the colorations correctly represent the intensities of compounds in general, and according to the coloration we can infer which compound is the majority in each part of the plant].

Comments 4: I would recommend the authors to put Tables 2 and 3 prior to the heatmaps.]

Response 4: [We thank you for your excellent recommendation, it was considered and Tables 2 and 3 were placed before the Heatmaps.]

Comments 5: [Figure 3 should be replaced with a better resolution figure.]

Response 5: [Thank you very much for the recommendation, figure 3 has been changed to have a better resolution as you suggested.]

Comments 6: [Table 2 – replace Uridin with Uridine.]

Response 6: [The name of the compound has been corrected, and it has been verified that it is correct throughout the document.]

Comments 7: [Figure 4 should be replaced with a better resolution figure.]

Response 7: [Thank you very much for the recommendation, Figure 4 was modified to have better resolution as you suggested.]

Comments 8: [In the Discussion part the authors could compare their results obtained by UHPLC with those of their previous work (ref. 2).]

Response 8: [Thank you very much for your recommendation, it enhances the results obtained in the present investigation. The suggested information has been added to the discussion section (Lines 246-253).

[“In contrast to the previous analysis [2] of soxhlet extracts of leaves and pseudobulbs of P. karwinskii collected from churches in Zaachila Oaxaca, in the present study, more compounds were identified for each part of the plant. This variation could be due to the method used for the extraction of the compounds, as well as to the loss or degradation of compounds due to the time they remained on the altars of the churches because of their ceremonial use and the manipulation of the plants. In the previous study, the origin of the plant was not considered, as they had been collected from different parts of the state for ceremonial purposes”]

Reviewer 2 Report

Comments and Suggestions for Authors

Dear Authors

After a review of your manuscript, I am sending you the following comments and suggestions for improving your work.

Manuscript Evaluation Report

General Comments

The manuscript titled "Variability in the chemical composition of leaves and pseudobulbs among localities of Prosthechea karwinskii (Orchidaceae) in Oaxaca, Mexico" presents a solid and novel investigation into the chemical variations of an endemic orchid species. The general structure complies with the standards of scientific publications, although there are areas for improvement in writing, clarity of some sections, and adaptation to the format of the journal Plants. The analysis is relevant to the fields of natural product chemistry and plant conservation.

Detailed Comments

TITLE

Concise and descriptive. Recommendation: Consider whether it can be further simplified without losing key information.

ABSTRACT

It presents the objectives, methods, and main results. Areas for improvement: 1) Add details about the sample size. 2) Review whether the value of the novel contribution can be included in a more impactful final sentence.

INTRODUCTION

Provides an adequate context and theoretical justification. Areas of improvement: 1) Include more up-to-date references on chemical variability and its relationship with environmental factors. 2) To better clarify the specific hypothesis of the study. 3) Consolidate the language to avoid repetitions.

MATERIAL AND METHODS

It adequately details the techniques of extraction and chemical analysis. Areas of improvement: 1) Include a diagram or outline of the extraction process to facilitate understanding. 2) Ensure that all UHPLC parameters are described according to reproducible guides. 3) Include more information on the controls and replications performed. 4) Specify in greater detail the collection and research permits granted by the corresponding authorities, including registration numbers or authorisation codes. 5) Ensure that the taxonomic classification provided by a botanical authority is described and confirm in which herbarium the samples are stored.

RESULTS

Well-structured, supported by clear tables and figures. Areas of improvement: 1) Integrate more statistical analysis into tables and figures, highlighting significant differences. 2) Add a brief description in the text that better connects the figures with the results discussed.

DISCUSSION

Relate the results with previous studies. Areas of improvement: 1) Expand the discussion of how specific environmental factors influence the identified metabolites. 2) Propose clearer hypotheses for future studies based on the findings.

CONCLUSION

It answers the question posed and highlights the practical implications. Areas for improvement: 1) Reformulate to avoid extrapolations that are not directly supported by the results.

REFERENCES

Adequate number of relevant references. Areas for improvement: 1) Review formatting agreement with Plants standards. 2) Ensure that all in-text citations are listed in references and viceversa.

Recommendations

1) Review the structure of the manuscript to fully align with the Plants guidelines. 2) Improve the clarity of the text, especially in the Introduction and Discussion. 3) Include more robust and detailed statistical analyses. 4) Ensure consistency in the format of references and figures. 5) Refer to the journal's instructions for authors and make adjustments to the length of the manuscript if necessary. 6) Clearly specify the collection and research permits, the taxonomic classification performed, and the herbarium where the collected samples are stored.

Strengths and Weaknesses

1) Novel research, good data organisation, relevant contribution to the conservation and use of medicinal plants. 2) Lack of greater depth in data analysis and connection between results and discussion.

Publishability

The manuscript is potentially publishable in Plants but requires minor modifications to fully meet the standards of scientific quality and editorial format.

Yours sincerely,

Reviewer

Author Response

RESPONSES TO REVIEWER’S COMMENTS

On behalf of all the authors, we sincerely appreciate the time and effort dedicated to reviewing this manuscript. Your feedback was invaluable in enhancing its quality. We have carefully considered your comments and made the corresponding revisions in the corrected manuscript. In the corrected version, the changes are highlighted in yellow, allowing you to see the modifications made. Additionally, we have provided a point.by- point response to each comment.

REVIEWER 2

Comments 1: [TITLE, Concise and descriptive. Recommendation: Consider whether it can be further simplified without losing key information.]

Response 1: [We appreciate your recommendation, it was considered, and we simplified the title without losing the key information, the new name is on line 2 of the document, “Chemical variation of leaves and pseudobulbs in Prosthechea karwinskii (Orchidaceae) in Oaxaca, Mexico.”]

Comments 2: [ABSTRACT, It presents the objectives, methods, and main results. Areas for improvement: 1) Add details about the sample size. 2) Review whether the value of the novel contribution can be included in a more impactful final sentence.]

Response 2: [We appreciate very much the recommendation for the improvement of our work.

Point 1) Details regarding sample size were added to the abstract of the manuscript. The information can be found on line 18 of the document. “95-790 and 376-3,900 g of fresh material for leaves and pseudobulbs, respectively, per locality”.

Point 2) a final sentence has been added to the abstract to address this reviewer's comment.]

Comments 3: [INTRODUCTION, Provides an adequate context and theoretical justification. Areas of improvement: 1) Include more up-to-date references on chemical variability and its relationship with environmental factors. 2) To better clarify the specific hypothesis of the study. 3) Consolidate the language to avoid repetitions.]

Response 3: [We appreciate the recommendations made to improve the introduction. In response, we have made the following changes:

Point 1) We updated the references on chemical variation and its relationship to environmental factors, Lines 57-67 and references 9-12 updated.

Point 2) In the last paragraph of the introduction, information was included to address the reviewer's comment to improve the study's hypothesis. Lines 68-82.

Point 3) We consolidated the language and wording to avoid repetitions.]

Comments 4: [MATERIAL AND METHODS, It adequately details the techniques of extraction and chemical analysis. Areas of improvement: 1) Include a diagram or outline of the extraction process to facilitate understanding. 2) Ensure that all UHPLC parameters are described according to reproducible guides. 3) Include more information on the controls and replications performed. 4) Specify in greater detail the collection and research permits granted by the corresponding authorities, including registration numbers or authorization codes. 5) Ensure that the taxonomic classification provided by a botanical authority is described and confirm in which herbarium the samples are stored.]

Response 4: [We appreciate the recommendations made to improve the materials and methods section. We made the following changes:

Point 1) A scheme of the extraction process was included for ease of understanding (Figure 6) on line 308 in the materials and methods section.

Point 2) More details about UHPLC-ESI-qTOF-MS/MS analysis were added to ensure reproducibility, lines 322-325.

Point 3) The results corresponding to the extraction yields are the average of three replicates, line 104 and Table 1. However, it was not possible for us to perform replicates of the UHPLC-ESI-qTOF-MS/MS analyses, so the relative intensity values in Tables 2 and 3 have no replicates.

Point 4) In the Materials and Methods section, information about the collection permit for the plant samples was included. In Table 1, information on the voucher specimens and the herbarium collection where they were deposited was added.

Point 5) In the Materials and Methods section, it is described who validated the taxonomic identity of the species and the classification system used to assign the scientific name.]

Comments 5: [RESULTS, Well-structured, supported by clear tables and figures. Areas of improvement: 1) Integrate more statistical analysis into tables and figures, highlighting significant differences. 2) Add a brief description in the text that better connects the figures with the results discussed.]

Response 5: [We are very grateful for recommendations to improve the results section.

Point 1) Unfortunately, it was not possible for us to perform replicates of the UHPLC-ESI-qTOF-MS/MS analysis, which prevents us from adding a more detailed statistical analysis, but we will consider it for future research derived from this work.

Point 2) The description was added in the text to better connect the figures with the results and discussion]

Comments 6: [DISCUSSION, Relate the results with previous studies. Areas of improvement: 1) Expand the discussion of how specific environmental factors influence the identified metabolites. 2) Propose clearer hypotheses for future studies based on the findings.]

Response 6: [We appreciate the recommendations to improve the discussion section. In response we made the following changes:

Point 1) We expand the discussion on the influence of environmental factors on the identified metabolites, lines 230-236.

Point 2) According to the results obtained we propose research questions to be answered in future studies based on our finding, lines 257-266, 274-275.]

Comments 7: [CONCLUSION, It answers the question posed and highlights the practical implications. Areas for improvement: 1) Reformulate to avoid extrapolations that are not directly supported by the results.]

Response 7: [We appreciate the recommendation to improve the conclusions section. In response, we have expanded the conclusions section to include the information requested by the reviewer, lines 357-363]

Comments 8: [REFERENCES, Adequate number of relevant references. Areas for improvement: 1) Review formatting agreement with Plants standards. 2) Ensure that all in-text citations are listed in references and viceversa.]

Response 8: [We thank you very much for the recommendations for the references section, we agree with them and make the following changes:

Point a) We corrected the format of references according to the requirements of Plants journal.

Point b) We checked all references to make sure that there were no references that were not in the paper and vice versa.]

Comments 9: [Recommendations

1) Review the structure of the manuscript to fully align with the Plants guidelines. 2) Improve the clarity of the text, especially in the Introduction and Discussion. 3) Include more robust and detailed statistical analyses. 4) Ensure consistency in the format of references and figures. 5) Refer to the journal's instructions for authors and make adjustments to the length of the manuscript if necessary. 6) Clearly specify the collection and research permits, the taxonomic classification performed, and the herbarium where the collected samples are stored.]

Response 9:

[We are grateful for all the recommendations provided, they really helped to improve the quality of our manuscript. Below we list the changes made according to these points.

Point 1) We checked that the structure of the manuscript was adjusted to the requirements of Plants journal.

Point 2) We improved the quality of the introduction and discussion sections as indicated above.

Item 3) Unfortunately, it was not possible for us to perform replicates of the UHPLC-ESI-qTOF-MS/MS analysis, the lack of replicates prevents us from adding a more detailed statistical analysis, but we will consider it for future research derived from this work.

Item 4) We corrected the references and figures.

Item 5) We have adjusted the manuscript to ensure its format complies with the journal's guidelines.

Item 6) We added information corresponding to the collection permits and taxonomic classification made in the section of materials and methods (Lines 279-280, 287-289), the information corresponding to the herbarium where the samples are stored was added to Table 1 (line 104).]
